# Rapidly sequence-controlled electrosynthesis of organometallic polymers

Jian Zhang[1,2], Jinxin Wang[1,3], Chang Wei[1,3], Yanfang Wang[1], Guanyu Xie[1,3], Yongfang Li[1,3] & Mao Li [1,2]✉

Single rich-stimuli-responsive organometallic polymers are considered to be the candidate for ultrahigh information storage and anti-counterfeiting security. However, their controllable synthesis has been an unsolved challenge. Here, we report the rapidly sequence-controlled electrosynthesis of organometallic polymers with exquisite insertion of multiple and distinct monomers. Electrosynthesis relies on the use of oxidative and reductive C–C couplings with the respective reaction time of 1 min. Single-monomer-precision propagation does not need protecting and deprotecting steps used in solid-phase synthesis, while enabling the uniform synthesis and sequence-defined possibilities monitored by both UV–vis spectra and cyclic voltammetry. Highly efficient electrosynthesis possessing potentially automated production can incorporate an amount of available metal and ligand species into a single organometallic polymer with complex architectures and functional versatility, which is proposed to have ultrahigh information storage and anti-counterfeiting security with low-cost coding and decoding processes at the single organometallic polymer level.

[1] State Key Laboratory of Polymer Physics and Chemistry, Changchun Institute of Applied Chemistry, Chinese Academy of Sciences, Changchun 130022, China. [2] University of Chinese Academy of Sciences, Beijing 100049, China. [3] University of Science and Technology of China, Hefei 230026, China. ✉email: limao@ciac.ac.cn

Precise insertion of multiple and distinct monomers into an artificial single polymer according to a desirable pattern is one of the biggest challenges for the synthesis of digital macromolecules[1–8]. Ideal synthesis should be reproducible, general, and low cost to provide the polymers with highly exquisite control over the insertion of multiple monomers[9]. Iterative synthesis, as a truly sequence-controlled polymerization of multiple monomers, can be divided into liquid-phase and solid-phase synthesis[7]. Each monomer is added at a time to the end of a growing macromolecular chain, then reaction debris is separated from the chain. These steps are repeated for next monomer in the sequence. Liquid-phase synthesis allows straightforward sampling for analysis with requirement of tedious chromatographic purification and time-consuming solution evaporation[10,11]. Solid-phase synthesis[12] plays an irreplaceable role as one of the most successful methods because of its simple purification and easy automation[13]. However, solid-phase synthesis has been hampered by overall efficiency and poor atom economy[10,14–17]. Though the liquid-phase and solid-phase synthesis can theoretically be extended to many kinds of monomers, this approach remains extremely difficult in making complex molecular architectures and tuning functional versatility, and precludes itself for synthesis and applications of materials. Making the connection in opposite direction between organic synthesis and electrochemistry has been realized to hold significant potential in both areas[18]. For example, the oxidative and reductive reactions in general organic synthesis are incompatible in identical solution, but electrochemical stimuli can enable switchable oxidative and reductive reactions to take place at the interface of identical electrode. The reaction at solid–liquid interface can be accelerated by controlling potential and current intensities, while the electrochemistry can simplify the purification and post-treatment processes[19,20].

In this paper, we describe the marriage of solid-phase synthesis with electrochemistry to rapidly generate uniform and sequence-controlled organometallic polymers with exquisite control over the composition and sequence of polymer backbone. Selected reactions of iterative electrosynthesis are oxidative self-coupling of N-phenyl carbazolyls and reductive self-coupling of vinyls on pyridines coordinated with metal cores between two monomers (Fig. 1a). Electrosynthesis initiates by the oxidative coupling of carbazolyls between the monomer in solution and pendant of complex self-assembled on ITO coated glass (Supplementary Figs. 1–4), and propagates via switching reductive and oxidative couplings in alternative solutions containing the same or different monomers (Fig. 1b). For ideal synthesis in single-monomer precision, precluding the possible coupling reaction of organometallic polymers with the dimer existed in solution from self-coupling of monomers, the oxidative and reductive reactions of identical monomer should be conducted in their individual solution. Rapid iterative electrosynthesis requires only 1 min or less time for each addition of monomers. Multinary monomers can be subsequently encrypted into sequence-controlled organometallic polymers.

## Results

### Iterative electrosynthesis of homo-organometallic polymers.

For iterative electrosynthesis of organometallic polymers (Fig. 2, Supplementary Figs. 5–6), on $M^{II}PX$ self-assembled ITO (indium tin oxide) coated glass, coupling reactions of carbazolyls and vinyls in 0.5 mM $M^{II}XY$ after optimizing experimental condition were conducted by cyclic voltammetry (CV) at oxidative potential range ($E = -0.50$–$1.0$ V, 50 mV s$^{-1}$, 1 cycle, 1 min) and reductive potential range ($E = -0.50$ to $-1.8$ V, 50 mV s$^{-1}$, 1 cycle, 52 s), respectively. Both reactions were well studied by Bard's[21], our[22] and worldwide groups[23–25]. The real-time optical and electrochemical monitoring can be used to ensure completely coupling proceed of each addition of monomer. As shown in Fig. 2a–d, the organometallic polymers grow in single-monomer precision, demonstrated by single-monomer-dependent absorption intensities (at 505 and 680 nm) and current intensities of redox peaks ($Os^{2+/3+}$, $E_{1/2} = 0.56$ V vs. Ag/Ag$^+$), according to single-monomer self-assembled. As the number of Os units increases with respect to single-monomer self-assembled, the absorption values and current intensities of organometallic polymers, based on random data of UV–vis spectra and statistical data of CV measurements on entire substrate, increase regularly and exhibit an excellent linear relationship ($R^2 = 0.991$–$0.998$) with switching times of oxidative and reductive reactions, indicating the polydispersity of organometallic polymers could be ignored. Thus, the single-monomer addition and the length of organometallic polymers can be well controlled towards quantitative and uniform synthesis. Here, the electrochemical cell was open to air with simple argon bubble, and this synthesis of organometallic polymers could be further optimized by the reaction time and frequency for each step towards further ideal synthesis. Electrosynthesis is rapid (1 min for each addition), and possibly independent of metal species and organic ligands of complexes, thus they provide significantly high controllability toward uniform synthesis, compared to well-known method based on coordination chemistry. To date, the iterative synthesis of organometallic polymers were mostly achieved by the metal coordination between almost Fe$^{2+}$ and terpyridine on solid substrate, which took up to 24 h for each addition of single monomer at room temperature[26–28]. Absorption laying at 505 nm is attributed to metal-to-ligand [Os$^{II}$(d$\pi$) to tpy($\pi^*$)] charge transfer (MLCT) transition, while the additional band at 680 nm seems to be owing to spin-forbidden MLCT transition from $^1$[Os$^{II}$(d$\pi$)$^6$] to $^3$[Os$^{II}$(d$\pi$)$^5$tpy($\pi^*$)$^1$][29]. Iterative reaction for tenth addition of monomers becomes time consuming because the length of organometallic polymers standing on the electrode surface has reached the width of the electric double layer, which is generally considered to have the thickness of ~20 nm from the surface of electrode. As auxiliary evidence, the resulting organometallic polymers on electrode have a thickness of 19.7 nm (Supplementary Fig. 7), which is in good agreement with the theoretical value of 20.3 nm in the extended state of organometallic polymer. Height differences of organometallic polymers observed in AFM images during iterative synthesis are less than length of 2.0 nm monomer (Supplementary Fig. 8), in good agreement with iterative addition in single-monomer precision. Long polymers with over 10 units could be synthesized and grow along the ITO surface without the limitation of electric double layer if the organometallic polymers were well separated into isolated state lying on the ITO surface.

This real-time monitoring, uniform and reproducible synthesis enables these organometallic polymers with good electrochemical stability (Supplementary Fig. 9) to be reliable for digital storage and reading of molecular information. In the first proof of concept, organometallic polymers containing various lengths and species of metal cores with distinguished optical and electric properties permit to express their own message, which can be easily deciphered by the X-axis position (metal species) and the Y-axis intensities (number of metal cores) of UV–vis absorption peaks and CV (Fig. 2). In principle, a wide species of metal cores and organic ligands with electrochemically stable C-metal and N-metal coordination bonds can be incorporated into organo-homo-metallic polymers to vary the molecular information with desirably electrochemical and optical features at area-selective positions with nano- or micro-scale of patterned electrodes, which could individually work as area-selective devices[30].

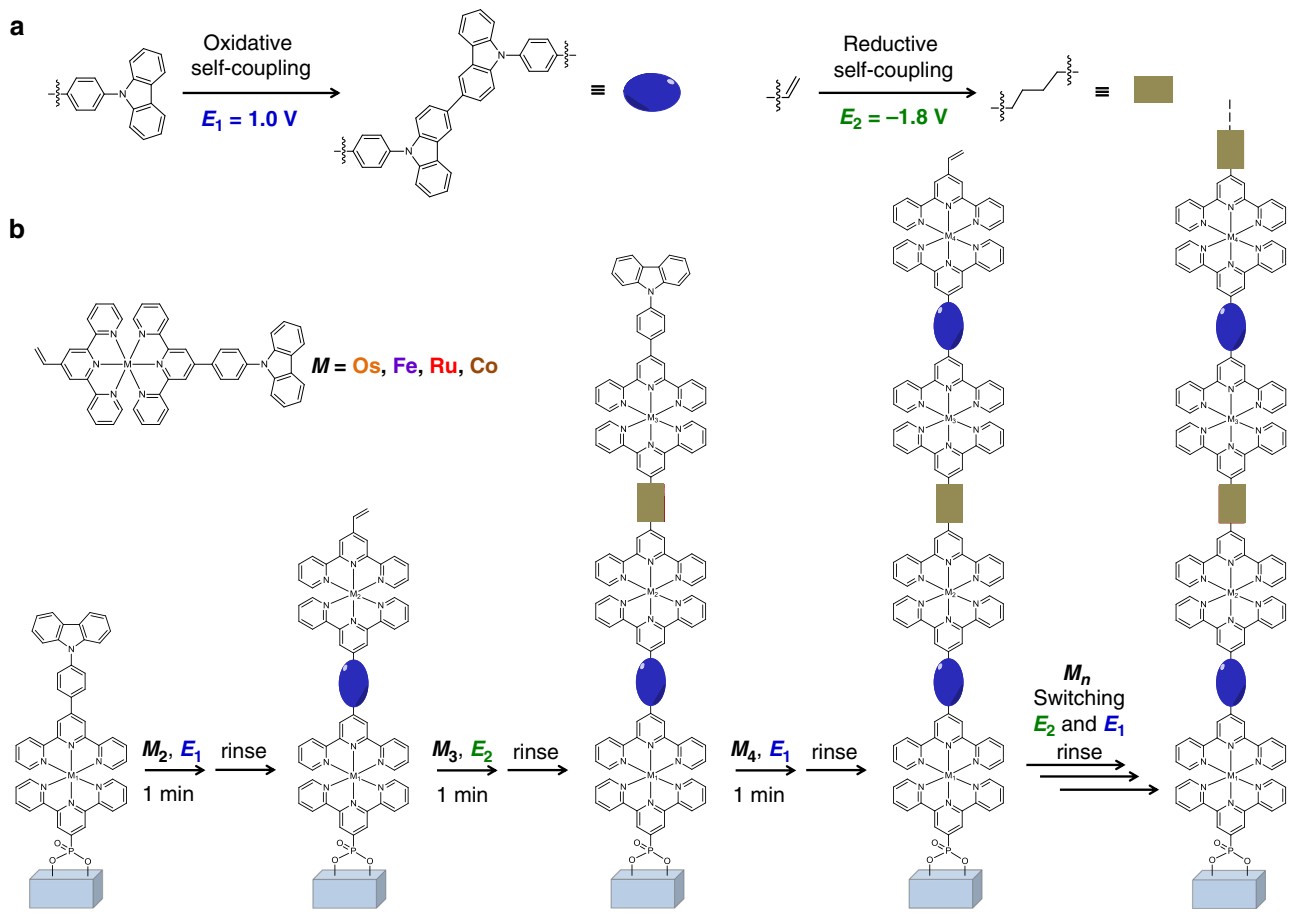

**Fig. 1 Rapidly sequence-controlled electrosynthesis of organometallic polymers. a** Illustrations of electrochemical oxidative and reductive couplings for iterative synthesis at 1.0 V and −1.8 V, respectively, on ITO coated glass. **b** Illustration of iterative synthesis of sequence-controlled organometallic polymers by switching oxidative and reductive reactions of distinct monomers.

**Iterative electrosynthesis of heter-organometallic polymers**. Iterative synthesis of $M_1^{II}XY$ on $M_2^{II}PX$ ($M_2 \neq M_1$) monolayer self-assembled on ITO glass was also feasible (Fig. 3, Supplementary Fig. 10), indicating X–X and Y–Y couplings are independent on species of metal cores of $M^{II}XY$. Three binary sequence-coded organometallic polymers of Os and Fe were synthesized (Fig. 3, Supplementary Figs. 11–12). As shown in Fig. 3a–d, the absorption peaks at 576 nm (metal-to-ligand [$Fe^{II}$ ($d\pi$) to tpy($\pi^*$)] charge transfer (MLCT) transition)[31] and current intensities of redox peaks (p1 and p2) rise up regularly as the function of switching times of oxidative and reductive reactions, indicating a successful iterative synthesis. The increase in absorption intensities at 505 nm can be also observed owing to iterative addition of shoulder absorption intensities of Fe units. Each iterative addition is easily observed in Fig. 3d that the current intensities are jumping up with the increase of Os or Fe units, while the current intensities of Fe or Os units remain stable. The single-monomer changes of current intensities can be easily distinguished, indicating a reliably sequence-controlled alternative addition in single-monomer precision. These observations can be also obtained in other binary (Fig. 3e–h) and ternary (Fig. 4, Supplementary Figs. 13–16) sequence-controlled organometallic polymers. Therefore, in the second proof of concept of molecular information storage, the coding of organometallic polymers can be also recognized by redox peaks of metal number and species in corresponding polymers. In these cases, the organometallic polymers with possibly different structures as impurities during iterative can be ignored. Importantly, the similar current densities (or cover area) of redox peaks (p1 and p2) of binary and ternary organometallic polymers in Fig. 3c, and Supplementary Figs. 11c, 12c should significantly indicate the same iterative additions of two species of metal cores. It implies that the coupling conversion does not have obvious change during iterative synthesis of different monomers. Compared to p1 and p2, the p3 shows a large redox peak because of the partial overlap with redox peak of 3,3′-bicarbazolyl units. The structural characterizations of organometallic polymers assembled on solid surface have been a challenge[32,33]. We found that the dimer already got worse solubility with time and gradually precipitated from the solution (Supplementary Figs. 17–19), these organometallic polymers with molecular weight of 7–13 kDa and length of 20 nm are difficult to be examined by solution processed NMR, Mass and GPC because the sample of 5 mg for structural characterizations does need the electrode of 5000 cm². Here the partial fragments of organometallic polymers can be found by Mass spectra on ITO surfaces (Supplementary Figs. 20–28). We have tried to obtain STEM image to analyze a periodically atomic structure in single organometallic polymer. Finally, the dark and bright atomic clusters of Os and Ru were observed. The difficulty of this experiment is sample preparation and transfer, while the structural formation change of film after sample preparation remains unknown (Supplementary Fig. 29). Usual chemical coordination for synthesis of organometallic wires[26–28] requires the high-quality solvents without external ions for synthesis and purification, and its controllability and reproducibility still remain stagnant and challenge in general metal and ligand species, and

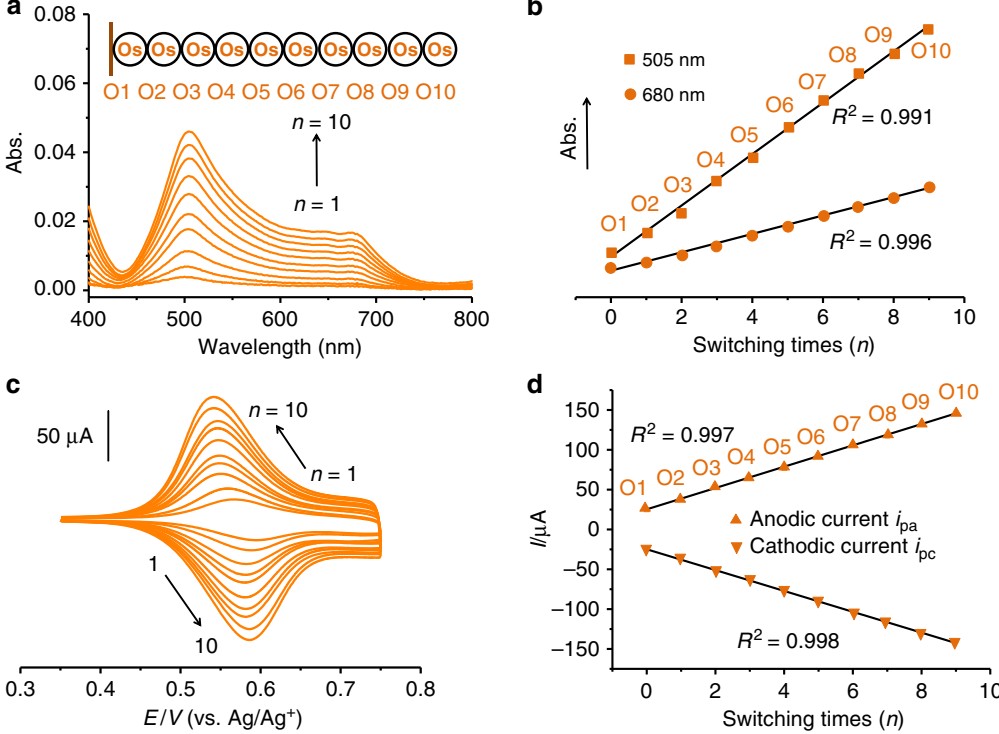

**Fig. 2 Electrosynthesis of homo-organometallic polymers. a, b** Illustrations of UV–vis absorbance and intensities as a function of switching times during iterative synthesis. **c, d** Illustrations of CV and its intensities of redox peaks as a function of switching times during iterative synthesis.

following coordination types for further sequence-controlled synthesis. Each addition of monomer for stepwise chemical coordination generally took very long time up to 24 h, which could cause the depolymerization and exchange of ligands and metal cores during the synthesis of organometallic polymers[34]. Here electrosynthesis not only shows the rapid synthesis and general potential for a lot of types of complexes with different species of metals and organic ligands, but also provides a high controllability in sequence-controlled synthesis of organometallic polymers.

As shown in Fig. 5a–c, a series of absorption spectra were collected from homo- and hetero-organometallic polymers with different compositions and sequences at potentials ranging from 0.55 to 1.05 V vs. Ag/Ag$^+$. Absorption peak of Os$^{II}$ homo-organometallic polymer at 495 nm changes at applied potentials from 0.55 to 0.75 V because of conversion of Os$^{II}$ to Os$^{III}$, and remains stable while applied potential climbs over 0.75 V. For hetero-organometallic polymers of Os$^{II}$ and Fe$^{II}$, the absorption peaks at 576 nm attributed to Fe complex units slightly drop down from 0.55 to 0.75 V, and dramatically decrease at applied potentials of >0.75 V because of conversion of Fe$^{II}$ to Fe$^{III}$. The absorption change of hetero-organometallic polymers at 576 nm between 0.55 and 0.75 V could be considered from the absorption shoulder change of Os$^{II}$ complex units. Thus, beside individual UV–vis spectra or CV measurement, the in situ optoelectrochemical data also enables us to read the number and species of metal cores of organometallic polymers, which could work in electrochromic devices.

**Sequence decoding of organometallic polymers.** Regarding the sequence decoding of organometallic polymers, establishing a library of spectra database of organometallic polymers with different sequences could help, because three binary and three ternary organometallic polymers can be recognized on detailed

comparisons of their UV–vis and fluorescence spectra (Fig. 5d, e). Photoluminescence from ligands excited by 310 nm shows the change from shoulder companied peak to sharp peak probably owing to the change of metal cores at two ends of ligands. It is well known that the absorption peak and redox peak of complex can be altered by modifying or changing organic ligands of complexes. As an alternative method for sequence decoding of organometallic polymers in future, the sequence decoding will become easy if 10 kinds of complexes containing the identical metal core and different organic ligands with their own optical and electric features were used for first to tenth particular positions in single organometallic polymer during iterative synthesis. Therefore, the sequence decoding of organometallic polymers will be possible via the further design of monomers used in this study. In this case, the sequence decoding of ternary organometallic polymers needs 30 kinds of monomers.

**Information storage and anti-counterfeiting security.** The highly controlled optical properties by sequencing organometallic polymers (Fig. 5d, e) offer a truly opportunity to determine quantitative structure-property relationships for designing materials. In principle, an amount of available metal cores in periodic table of elements including Co (Supplementary Fig. 30), Ir, etc.[35,36] could be incorporated into organometallic polymers. Generally, every monomer in digital polymer expresses 0- or 1-bit, and polymer with eight monomers gives a letter in ASCII (American Standard Code for Information Interchange)[37]. In this paper, every monomer (metal core) can express one or two (c.a. Ru for r or ru) letters, and single organometallic polymer could express a word or sentence because of a number of available ligands and metal species with rich-responsive states by electric or/and optical stimuli. Therefore, these organometallic polymers theoretically have the ultrahigh information storage with exponential enhancement compared with single kind of metal cores[38].

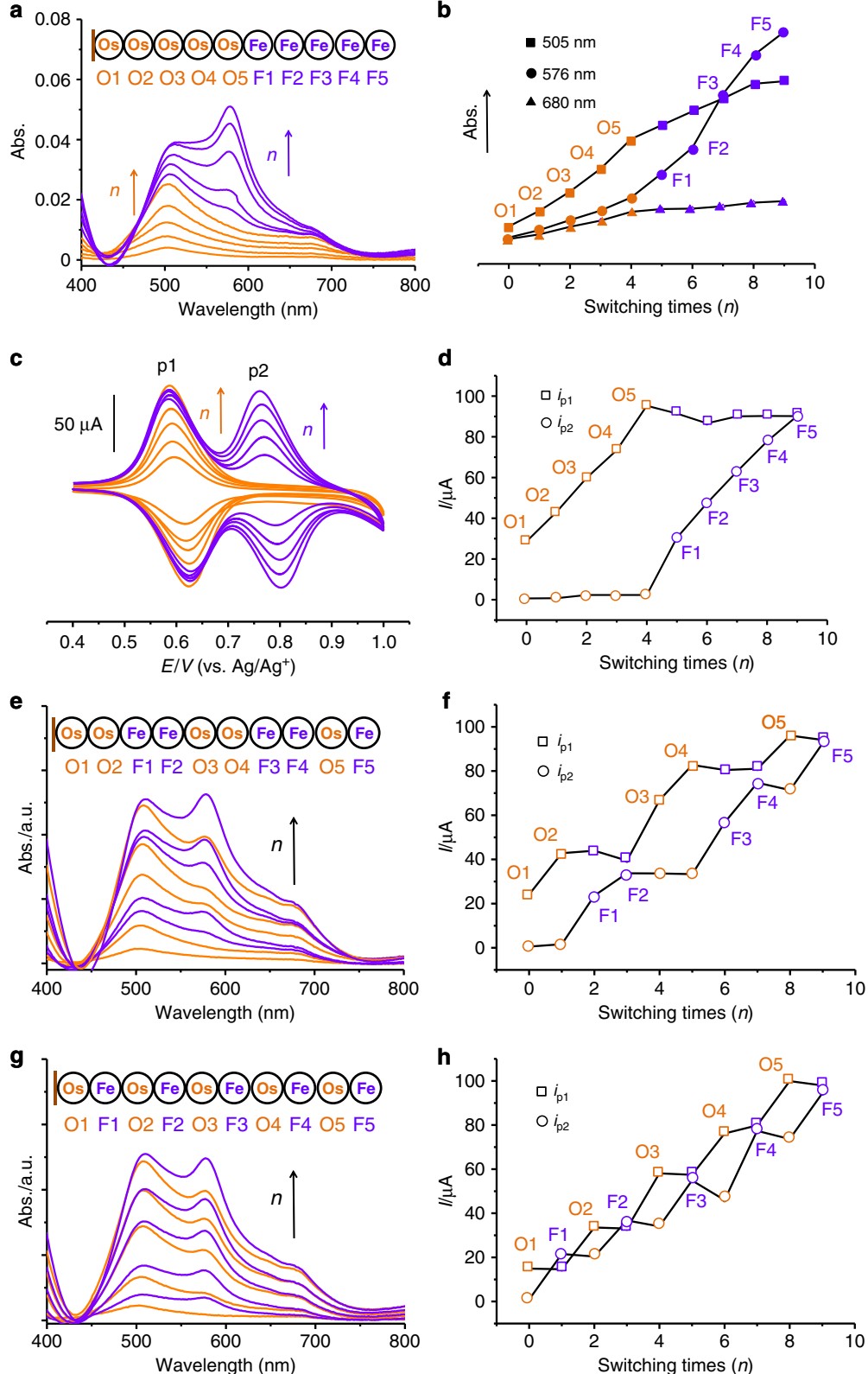

**Fig. 3 Electrosynthesis of binary organometallic polymers. a**, **b** Illustrations of UV–vis absorbance and intensities as a function of switching times during iterative synthesis. **c**, **d** Illustrations of current intensities of oxidative peaks as a function of switching times during iterative synthesis. **e–h** Illustrations of UV–vis absorbances and current intensities of oxidative peaks as a function of switching times during iterative of organometallic polymers with different sequences.

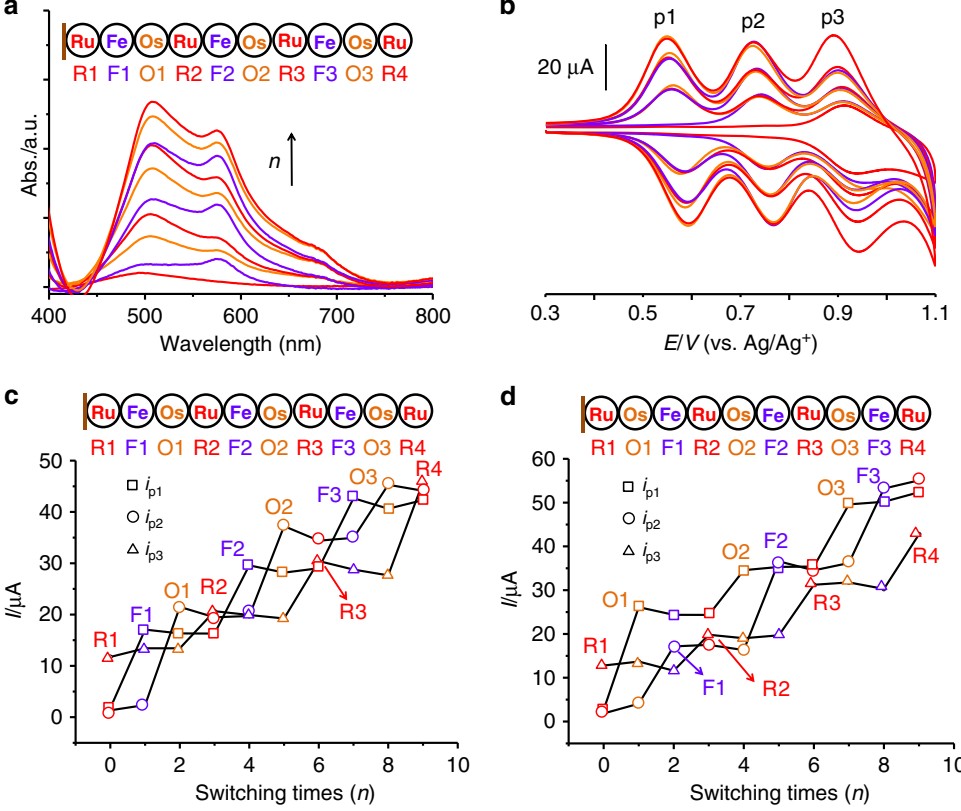

**Fig. 4 Electrosynthesis of ternary organometallic polymers. a, b** Illustrations of UV–vis absorbance (left) and CV (right) as a function of switching times during iterative synthesis. **c, d** Illustrations of current intensities as a function of switching times during iterative synthesis of organometallic polymers with different sequences.

Ten and twenty kinds of monomers in single organometallic polymers containing 10 units with different electrochemical and optical features have the sequences of over $3.6×10^6$ and $6.7×10^{11}$. This large library of sequences with difficultly structural characterization ensures these organometallic polymers to have ultrahigh anti-counterfeiting security. The electrochemistry and UV–vis spectra are considered to be the convenient and low-cost coding and decoding processes for practical applications in both liquid solution and solid states compared to other techniques such as NMR and Mass spectra. In addition, this uniform synthesis facilitates thickness controlled fabrications in single-monomer precision on large area compared to conventional electropolymerization (Supplementary Fig. 31).

## Discussion

We have successfully demonstrated a rapid, general and low-cost production of uniform and sequence-controlled organometallic polymers with exquisite insertion of multiple and distinct monomers using bottom-up iterative electrosynthesis via stable C–C couplings. This sequence-controlled electrosynthesis offers a true opportunity to determine quantitative structure-property relationships for designing materials. Highly efficient reactions, simple washing purification and real-time monitoring ideally lend them to a potentially mechanized and automated process through the integration of computer controls to take the desirable molecular structures of a desired product and output[39–42]. The advances of organometallic polymers with tailored organic ligand, metal species and number have been further proposed for ultrahigh-density molecular coding with decoding solutions. Further potential applications may not necessarily require long

organometallic polymers. The possible electrodes can be metals (c.a. gold), some metallic oxides (c.a. fluorine-tin oxide, titanium dioxide) and carbon[43,44]. We envisage that these organometallic polymers with highly tunable structures, functions, and rich-stimuli-responsive switch as an excellent molecular platform will also hold great promise in applications, such as catalysis[45], electrochromics[46], molecular electronics[47], capacitances[48], etc. Studies toward these applications are currently taking place in our laboratory.

## Methods

**Materials and syntheses.** Materials and reagents used in this study were purchased and used without further purification. Solvents for chemical synthesis and electrochemical measurements were purified by distillation. All aqueous electrolyte solutions and water washings were performed with reagent grade deionized water. The synthetic routes and structural characterizations of the monomers are presented in supplementary Figs. 32–60.

**Fabrication of self-assembled monolayer.** Self-assembled monolayers used in this study were prepared by immersing freshly cleaned ITO coated glass in 0.1 mM methanol solution of Ru$^{II}$XP and Os$^{II}$XP for 12 h in darkness, followed by sonication cleaning in ethanol bath, then washed with CH$_2$Cl$_2$ and dried under N$_2$ flow.

**Electrochemical synthesis.** Electrochemical measurements were performed using a typical one-compartment, three-electrode setup. Supporting electrolytes Bu$_4$NPF$_6$ (tetrabutylammonium hexafluorophosphate) and Bu$_4$NClO$_4$ (tetrabutylammonium perchlorate) were dried for 24 h at 80 °C under vacuum before use. Ag/Ag$^+$ or Ag/AgCl (an AgCl coated Ag wire) were used as reference in organic and aqueous systems, respectively. ITO (8–12 Ω per □) and glassy carbon were used for working electrode. Pt wire served as counter electrode in all cases. Prior to electrochemical assembly, N$_2$ was purged to solution for 20 min. Iterative synthesis was conducted on self-assembled monolayers modified ITO as working electrode (working area: 1.0 cm$^2$) by alternatively applying positive and negative potential.

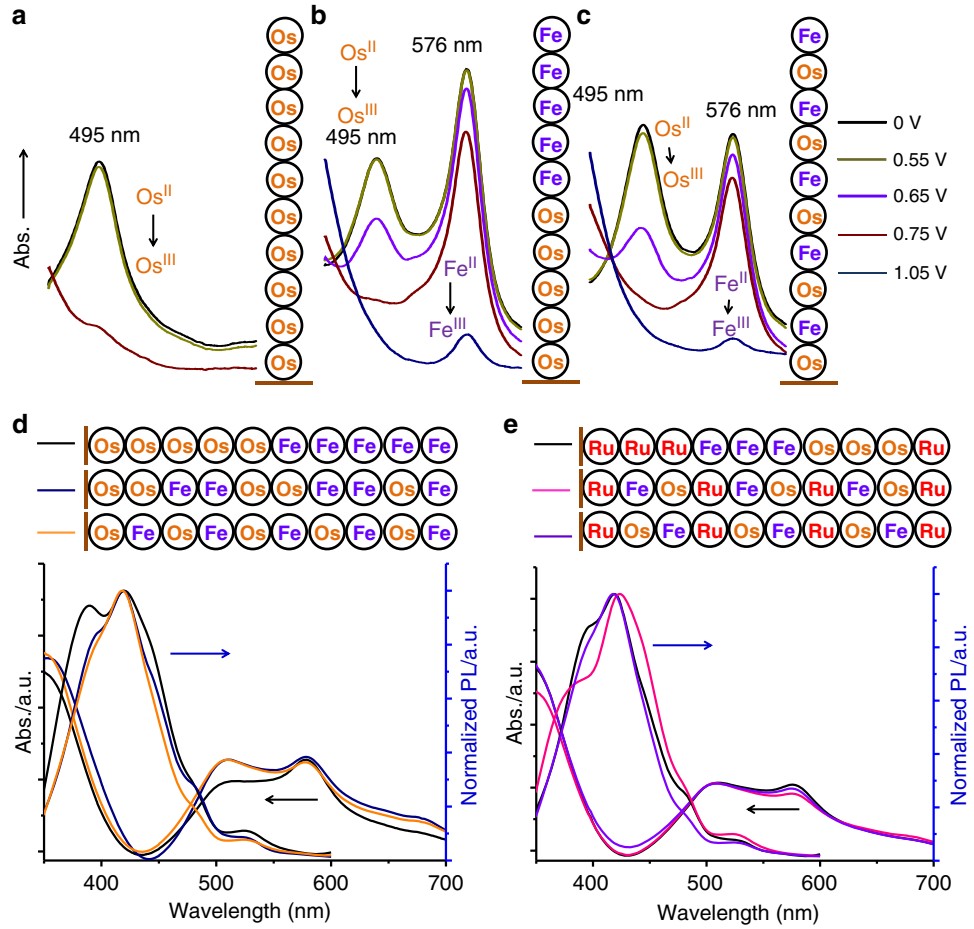

**Fig. 5 Distinct UV–vis absorption and photoluminescence spectra of organometallic polymers with different compositions and sequences.**
**a–c** Electrochromic behaviors at absorption range between 450 nm and 620 nm on ITO coated glasses at applied different potentials. **d, e** UV–vis absorption and photoluminescence (PL) spectra of binary and ternary organometallic polymers.

## Data availability

The source data underlying Figs. 2–5 and Supplementary Figs. 1, 3–5, 9–16, 30, 31 are provided as a Source Data file. All data are available from the corresponding author upon reasonable request.

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

## Acknowledgements

The authors are grateful to Prof. Guobao Xu, Prof. Zhaohui Su, Prof. Dongmei Cui and Prof. Youhua Tao (CIAC, CAS), Prof. Yu-Wu Zhong and Prof. Lang Jiang (IC, CAS), and Prof. Zhengbiao Zhang and Dr. Zhihao Huang (Soochow Univ.) for useful discussions. We thank Prof. Xiaopeng Li and Mr. Guangqiang Yin (Univ. South Florida) for Mass mesurements, Prof. Lin Gu and Dr. Qinghua Zhang (IP, CAS) for STEM mesurements, and Dr. Ying Lv (CIOMP, CAS) for fabrication of ITO coated silicon wafer. We also acknowledge the mass spectrometry characterization by Molecular Scale Lab. This work was supported by the National Natural Science Foundation of China (21774121, 91963125).

## Author contributions

M.L. designed and conceived this study, and wrote the manuscript. J.Z. synthesized all compounds and measured most of data. J.W., C.W., Y.W., G.X., and Y.L. were partially involved in completing the experiments.

## Competing interests

The authors declare no competing interests.
