## [Peer Review File · Nature Communications]

Reviewers' comments:

Reviewer #1 (Remarks to the Author):

This paper aims to demonstrate the synthesis of sequence-defined, coordination polymers through the use of successive, electrochemically induced oxidative/reductive couplings of carbazolyis and vinyl groups off of a solid surface. The authors sell this paper as an improved version of solid-phase synthesis that doesn't require expensive protecting groups and can be performed at faster speeds. The authors argue the sequence definition of the synthesized polymers could be used for coding molecular information. While these claims would merit an original and high impact paper, the evidence and implementation leave much to be desired.

The synthesized compounds cannot be considered polymers. They are far too short as they cannot be made with more than 10 units.

The terms "sequence-definition" and "monodisperse" need to be held up to greater standards of characterization. At bare minimum, there needs at least to be full NMR, MALDI, and GPC data demonstrating the unimolecularity of the synthesized "sequenced-defined polymers." While the associated absorbance spectra and CV provide some hints to some rough sequence control, it is far from sufficient in order for the authors to make their claims. If the authors are able to strongly demonstrate this is truly not possible with these coordination polymers, they need to at the very least demonstrate this using a nonmetal sequence-defined polymer.

To be compared to solid-phase synthesis, it needs to be possible to remove the molecules from their solid support and subsequently characterized. This has not been demonstrated.

If there's going to be a claim that these molecules can be used for molecular coding, there needs to be additional demonstration of the long term stability of the molecules.

Monodisperse is not a word. Use the words uniform or unimolecular alternatively.

The paper needs to be rewritten in a major way. There are far too many grammatical issues, improper diction, and stylistic gaffs.

Figure 1 needs to be redesigned to better the synthetic details in a more easily understandable fashion.

Unless the authors are able to address these concerns, the paper cannot live up to the claims it presents and is just another paper on surface modification.

Reviewer #2 (Remarks to the Author):

This is an excellent paper on the preparation of sequence defined polymers using electrochemistry. The paper is well organized and the data are well presented. The approach is very novel and I think it will be the first example of sequence defined polymers prepared via electrochemistry.

The authors have explained very well the concept in the introduction and placed their work in context. Perhaps, it will be also interesting to introduce some selective activation using visible light, which is a bit similar, for instance: *Angewandte Chemie International Edition* 56 (29), 8376-8383 or *Polymer Chemistry* 8 (32), 4637-4643.

In the results and discussion, it will be interesting to see additional characterizations, such as MALDI or ESI-MS or GPC chromatograms. This will provide useful information for the readers.

Reviewer #3 (Remarks to the Author):

This work by Li et al. describes the development of precision sequence control over coordination polymers using electrosynthesis. Characterization via UV-vis and CV confirm the single addition of a monomer during each cycle, with examples of homopolymers and copolymers of two and three different monomers (coordinated to different ions). The synthesis is rapid (~ 1 min per addition) and the characterization at each addition is compelling. Due to the width of the electric double layer, this work targets only oligomers (up to 10 monomers), but the authors argue that this provides a unique opportunity for data storage due to the unique absorption of different polymers. This work is elegant and of interest to a broad audience as the field of sequence-controlled polymers is expanding. Answers these questions will aid in identifying the significance of the work within the fields that are described.

1. The authors spend a lot of time describing the application to information storage; however, it is not clear from the data presented:

a. What is the resolution of this synthetic strategy? Is there a pathway envisioned towards synthesizing these polymers on a relevant substrate?

b. While there are a few absorption spectra for copolymers, there seems to be the implication that different sequences of the same monomers will have different absorption spectra, but there is no quantitative characterization of this, just a qualitative overlay. How will the sequences of the polymers be decoded using the absorption spectra?

c. In general, the authors make broad generalizations that these materials will provide "significant advantages of appealing information-related applications..." – more context as to the current state of the field with different (bio)polymer strategies would be valuable.

2. The comparison to solid-phase synthesis, and the inefficiency of using an excess of reagents that are washed away, is made multiple times. However, this synthetic strategy similarly exchanges the solution with every monomer addition. Are these solutions reused? Are lower concentrations of reagent required than solid-phase synthesis on a comparable surface?

3. Minor comments:

a. Figure 1 is hard to follow – can this be edited to be easier to follow for an external audience? Additionally, the aromatic ring structures have different lengths for the double bonds.

b. For the plots where the x-axis is labeled "switching time" – should this have a unit affiliated?

c. Some of the graphs appear to have a linear regression fit – can the R^2 value and any additional relevant information be provided?

Point-by-point response to the reviewer comments:

1

Reviewers' comments:

Reviewer #1 (Remarks to the Author):

This paper aims to demonstrate the synthesis of sequence-defined, coordination polymers through the use of successive, electrochemically induced oxidative/reductive couplings of carbazoyls and vinyl groups off of a solid surface. The authors sell this paper as an improved version of solid-phase synthesis that doesn't require expensive protecting groups and can be performed at faster speeds. The authors argue the sequence definition of the synthesized polymers could be used for coding molecular information. While these claims would merit an original and high impact paper, the evidence and implementation leave much to be desired.

The synthesized compounds cannot be considered polymers. They are far too short as they cannot be made with more than 10 units.

Response 1:

Owing to difficult synthesis of sequence-defined polymers with high molecular weight, this type of polymers usually does not have very long length of backbone. We thought that our molecular length of 10 units and molecular weights of 7~12 kDa could be acceptable, according to some examples: (1) Serpell, Sleiman, *Angew. Chem. Int. Ed.* 2014, 53, 4567. (2) Alabi, J. *Am. Chem. Soc.* 2014, 136, 13162. (3) Jamison, *PNAS* 2015, 112, 10617. (4) Barner-Kowollik, *Chem. Commun.* 2015, 51, 1799. (5) Konrad, Feist, *Nat. Commun.* 2016, 7, 13672. (6) Meier, *Angew. Chem. Int. Ed.* 2016, 55, 1204. (7) Zhang, *Nat. Commun.* 2019, 10, 1918. (8) Gao, *J. Am. Chem. Soc.* 2019, 141, 4541. (9) Xia, *Chem* 2019, 5, 2691.

We do understand that this concern came from rigorous considerations. Therefore, we have replaced “polymer” by “organometallic wire”.

Regarding “They are far too short as they cannot be made with more than 10 units.”

Our method could get longer backbone more than 10 units, but lose good controllability in our current experiments. In this paper, the length of monomer is ~2 nm, and the length of organometallic wire with 10 units has approached the width of electric double layer.

In order to overcome the limitation of electric double layer, we think that “long wires could be synthesized and grow along the ITO surface without the limitation of electric double layer if the organometallic wires were well-separated into isolated state lying on ITO surface”. We have added this explanation into our paper.

The terms "sequence-definition" and "monodisperse" need to be held up to greater standards of characterization. At bare minimum, there needs at least to be full NMR, MALDI, and GPC data demonstrating the unimolecularity of the synthesized "sequenced-defined polymers."

Response 2:

The structural characterizations of organometallic wires assembled on solid surface have been a challenge. (*Nat. Mater.* 8, 41-46 (2009). *Coord. Chem. Rev.* 346, 139-149 (2017).)

Regarding this concern, tiny amount of self-assembled organometallic wires and its solubility are key

issues. Mass spectrometry was utilized for analyzing the organometallic wires after treatment of HF, but no evidence was obtained. ITO samples are used in study have an assembled size of 1 cm^2 . Organometallic wires of 5 mg need the surface sample of 5000 cm^2 . Additionally, the solubility of these organometallic wires containing 10 metal cores remains unknown. Actually, after iterative synthesis of single organometallic wire with 10 repeated units, the precipitation can be found (new Figure S17), indicating the solubility of oligomer becomes worse, and also implying that the single organometallic wire with 10 repeated units will be difficult to be soluble. We have tried to obtain the dimer via electrolysis at 1.0 V on bare ITO, and found the dimer appeared as precipitation on electrode surface, which was demonstrated by NMR and Mass spectra (new Figure S18,19). We have added new Figure S17-19 into supporting information.

Figure S17. Photos of $\text{Ru}^{\text{II}}\text{XY}$ solutions before (a) and after (b) iterative synthesis, and photo of bare ITO electrode without pre-assembled molecule after electrolysis of $\text{Ru}^{\text{II}}\text{XY}$ at 1.0 V for 1h (c). The precipitation can be found after iterative synthesis, indicating the oligomers become insoluble, and also implying that the solubility of single organometallic wire with 10 units will be difficult. After electrolysis of $\text{Ru}^{\text{II}}\text{XY}$ at 1.0 V for 1h, the dark red film was found on ITO surface (c) and dissolved for measurements of NMR and Mass spectra as shown in Figure S18,19, demonstrating that this film was composed of dimer with bad solubility.

Figure S18. ^1H NMR spectra of Ru XY and its dimer in CD_3CN obtained by electrolysis in Figure 17c.

Figure S19. MALDI-TOF mass spectrum of $\text{Ru}^{\text{II}}\text{XY}$ dimer obtained by electrolysis in Figure 17c.

For measurements of our samples on ITO substrates:

In October 2019, Prof. Zhengbiao Zhang said that our sample could be measurable by their Mass spectra, but their staff refused our request in November 2019, because he seriously worried about glass falling into vacuum chamber, even the samples and glasses were processed by adequately milling ways. In Dec 2019, these samples (ITO size = $1\text{cm} \times 1\text{cm}$) were transferred to Prof. Xiaopeng Li in University of South Florida (<http://faculty.cas.usf.edu/xiaopengli1/Index.html>), who is

considered to have more experience in dealing with sample on solid substrate. Fe core was not incorporated into organometallic wires because its weak coordination is unstable for Mass measurements. In Jan 2020, new samples (ITO size = 2.5cm*7.5cm) were asked by Prof. Xiaopeng Li in order to match the accessories of instrument.

Here we have prepared 3 types of Ru and Os organometallic wires (111122222, 11221122112212, 1212121212, 1 = Ru, 2 = Os). Ru coordination is more stable than Os coordination in Figure S21-23. Mass does not reach the theoretical value probably due to weak coordination, which could lead more small fragments and also result in deviation in Ru and Os alternative organometallic wires, while there is extremely tiny wires on ITO substrate. Therefore, the alternative Ru and Os organometallic wires are not well-recognized in main peaks of their mass spectra (new Figure S24-28). In future, in order to obtain stable monomers, CNN or NCN types of monomers will replace NNN type of monomers in this paper.

In new Figure S21-23:

Molecular fragments of organometallic wire with theoretical mass of 2147.36 and 3513.20 are found in mass spectrum, and they can be clearly identified. These two fragments contain two Os cores and four Ru cores, respectively, indicating a diblock pattern of original copolymer. The broken coordination bonds between metal cores and ligands are deemed an obstacle to obtain the direct evidence of intact coordination polymer by mass spectrometry. Two fragments contain both structures of carbazolyl and vinyl dimerization, clarifying the elementary reactions of electrochemical iterative synthesis.

Figure S20. Organo-hetero-metallic wires of Os and Ru at ITO coated glass (2.5 cm×7.5 cm) for measurements of MALDI-TOF mass spectra. In the experiment, DCTB (20 mg/mL) was used as matrix and spotted on the polymers coated ITO (2.5 cm × 2 cm). The calibration was carried out using PMMA. Both reflection and linear modes were used to obtain signals. Fe core was not incorporated into organometallic wires because its weak coordination is unstable for Mass measurements. Herein, we have prepared 3 types of Ru and Os organometallic wires (111122222, 11221122112212, 1212121212, 1 = Ru, 2 = Os). Ru coordination is more stable than Os coordination in Figure S21-23. Mass does not reach the theoretical value probably due to weak coordination, which could lead more small fragments and also result in deviation in Ru and Os alternative organometallic wires, while there is only tiny wires on ITO substrate. Therefore, the alternative Ru and Os organometallic wires are not well-recognized in main peaks of their mass spectra (Figure

S24-28).

5

Figure S21. MALDI-TOF mass spectrum of Os and Ru organometallic wires in linear mode. Ru coordination is more stable than Os coordination.

Figure S22. MALDI-TOF mass spectrum of Os and Ru organometallic wires in reflection mode. Molecular fragments of organometallic wire with theoretical mass of 2147.36 and 3513.20 are found in mass spectrum, and they can be clearly identified. These two fragments contain two Os cores and four Ru cores, respectively, indicating a diblock pattern of original copolymer. The broken coordination bonds between metal cores and ligands are deemed an obstacle to obtain the direct evidence of intact coordination polymer by mass spectrometry. Two fragments contain both structures of carbazolyl and vinyl dimerization, clarifying the elementary reactions of electrochemical iterative synthesis.

Figure S23. MALDI-TOF mass spectrum of Os and Ru organometallic wires in reflection mode.

Figure S24. MALDI-TOF mass spectrum of Os and Ru organometallic wires in linear mode.

Figure S25. MALDI-TOF mass spectrum of Os and Ru organometallic wires in linear mode.

Figure S26. MALDI-TOF mass spectrum of Os and Ru organometallic wires in reflection mode.

Possible molecular fragments of # 2

Figure S27. MALDI-TOF mass spectrum of Os and Ru organometallic wires in linear mode.

Figure S28. MALDI-TOF mass spectrum of Os and Ru organometallic wires in reflection mode.

Possible molecular segments of # 3

With collaboration with Prof. Lin Gu and Dr. Qinghua Zhang in Institute of Physics, CAS, we have very recently in December tried to obtain STEM image to analyze an periodically atomic structure in single organometallic wire. The atomic clusters of Os and Ru with different brightness were observed. The difficulty of this experiment is sample preparation and transfer, while the structural formation change of organometallic wires after sample preparation and during observation remains unknown. We have added new Figure S31 into supporting information.

Figure S31. STEM image of Os^{II} and Ru^{II} alternative organometallic wire.

While the associated absorbance spectra and CV provide some hints to some rough sequence control, it is far from sufficient in order for the authors to make their claims.

Response 3:

As the number of metal core units increases with respect to self-assembled monolayer, the absorbance and current intensity of molecule increase regularly and exhibit a good linear relationship with switching times of potentials, based on **random data** of UV-vis spectra and **statistical data** of CV measurements on entire substrate. As shown in Figure 3, the single-monomer-changes in UV-vis spectra and CV can be easily distinguished, indicating that single-monomer-addition to organometallic wire is reliable.

In our paper, “statistical data of UV-vis spectra and CV measurements” has been changed to “random data of UV-vis spectra and statistical data of CV measurements on entire substrate”

If the authors are able to strongly demonstrate this is truly not possible with these coordination polymers, (**Response 2**) they need to at the very least demonstrate this using a nonmetal sequence-defined polymer.

Response 4:

This concern has been a great challenge, and also will be our future work after we understand how to well-characterize our organometallic wires.

The oxidative coupling reaction of carbazoles is independent during iterative synthesis, while the reductive coupling reaction of double bonds needs special functional units (metal core with bpy in our paper) to enhance the reactivity. We have previously predicted that the design of non-metal

compound is more difficult, because the reactivity of double bonds, the electrochemical stability in a wide range of oxidative and reductive potentials, and available monitoring of iterative synthesis should be seriously considered.

Synthesis of organometallic wires and their materials with hard solubility have been an unsolved challenge. This method therefore could be a very powerful tool to synthesize the organometallic wires, which could not be synthesized by general organic synthesis. Unfortunately, the characterizations (NMR, MALDI, GPC) of organometallic wires will also be hampered by hard solubility of these organometallic wires. This problem needs to be solved in future by unconventional characterization methods, for example, direct observation of grow processes of single organometallic wire as well as following paper. We need to redesign the molecule and fabricate the good substrate for this possibility.

Reprinted (adapted) with permission from J. Am. Chem. Soc. 2019, 141, 44, 17713-17720. Copyright (2019) American Chemical Society.

“Nanoribbons with nonalternant topology from fusion of polyazulene: carbon allotropes beyond graphene”, J. Am. Chem. Soc. 2019, 141, 17713.

Now, we are seeking the collaborators, which are experts at surface characterization.

To be compared to solid-phase synthesis, it needs to be possible to remove the molecules from their solid support and subsequently characterized. This has not been demonstrated.

Response 5:

This response is similar to Response 2.

If there's going to be a claim that these molecules can be used for molecular coding, there needs to be additional demonstration of the long term stability of the molecules.

Response 6:

We have added new Figure S9 into supporting information.

Figure S9. CVs from 1st to 1000th and the current intensities as function of CV cycles for self-assembled Os^{II} (a,b) and Ru^{II} (c,d) organometallic wires. The current intensities of redox peaks for both organometallic wires do not show significant change from 2nd to 1000th cycles in open air, indicating these organometallic wires have good electrochemical stability.

Monodisperse is not a word. Use the words uniform or unimolecular alternatively.

Response 7:

“Monodisperse” was changed to “uniform” in this paper.

The paper needs to be rewritten in a major way. There are far too many grammatical issues, improper diction, and stylistic gaffs.

Response 8:

The paper has been rewritten and polished very carefully.

Figure 1 needs to be redesigned to better the synthetic details in a more easily understandable fashion. Unless the authors are able to address these concerns, the paper cannot live up to the claims it presents and is just another paper on surface modification.

Response 9:

We have redesigned Figure 1 for easy understanding.

New Figure

Previous Figure

Reviewer #2 (Remarks to the Author):

This is an excellent paper on the preparation of sequence defined polymers using electro-chemistry. The paper is well organized and the data are well presented. The approach is very novel and I think it will be the first example of sequence defined polymers prepared via electrochemistry.

Response 10:

We do also think that this is the first example of sequence defined polymers prepared via electrochemistry.

We have added new Ref.19 and modified the sentence “This electrosynthesis as first sequence controlled electropolymerization¹⁹ offers a truly opportunity to determine quantitative structure–property relationships for designing materials.” in conclusion.

The authors have explained very well the concept in the introduction and placed their work in context. Perhaps, it will be also interesting to introduce some selective activation using visible light, which is a bit similar, for instance: *Angewandte Chemie International Edition* 56 (29), 8376-8383 or *Polymer Chemistry* 8 (32), 4637-4643.

Response 11:

“*Angewandte Chemie International Edition* 56 (29), 8376-8383.” was submitted in 2016, and “*Polymer Chemistry* 8 (32), 4637-4643.” was submitted in 2017. Therefore, we have added

“Angewandte Chemie International Edition 56 (29), 8376-8383.” into Ref.1.

14

In the results and discussion, it will be interesting to see additional characterizations, such as MALDI or ESI-MS or GPC chromatograms. This will provide useful information for the readers.

Response 12:

This response in details is similar to Response 2.

Reviewer #3 (Remarks to the Author):

This work by Li et al. describes the development of precision sequence control over coordination polymers using electrosynthesis. Characterization via UV-vis and CV confirm the single addition of a monomer during each cycle, with examples of homopolymers and copolymers of two and three different monomers (coordinated to different ions). The synthesis is rapid (~1 min per addition) and the characterization at each addition is compelling. Due to the width of the electric double layer, this work targets only oligomers (up to 10 monomers), but the authors argue that this provides a unique opportunity for data storage due to the unique absorption of different polymers. This work is elegant and of interest to a broad audience as the field of sequence-controlled polymers is expanding. Answers these questions will aid in identifying the significance of the work within the fields that are described.

1. The authors spend a lot of time describing the application to information storage; however, it is not clear from the data presented:

a. What is the resolution of this synthetic strategy? Is there a pathway envisioned towards synthesizing these polymers on a relevant substrate?

Response 13:

As the number of metal core units increases with respect to self-assembled monolayer, the absorbance and current intensity of molecule increase regularly and exhibit a good linear relationship **in single monomer precision**, based on **random data** of UV-vis spectra and **statistical data** of CV measurements on entire substrate, indicating a reliably quantitative production of uniform organometallic wires.

Possible substrates could be gold, FTO, TiO₂ and carbon (Marinescu, ACS Appl. Energy Mater. DOI: 10.1021/acsaem.8b01745; Lacroix, J. Am. Chem. Soc. 2012, 134, 5476). We have added these references as Ref. 21 and new sentence “the possible substrates are gold, FTO, TiO₂ and carbon.” into conclusion.

b. While there are a few absorption spectra for copolymers, there seems to be the implication that different sequences of the same monomers will have different absorption spectra, but there is no quantitative characterization of this, just a qualitative overlay. How will the sequences of the polymers be decoded using the absorption spectra?

Response 14:

For iterative synthesis of different monomers, the quantitative study on absorption spectra becomes difficult due to obvious overlaps of different absorption peaks. We may obtain the quantitative data if different absorption peaks could be well-separated in absorption spectra after the redesign of monomers. Herein, the current intensities in CVs of complexes become important for quantitative

study.

Regarding the sequence decoding of organometallic wires, there are two methods.

- (a) We can establish a library of spectra database of organometallic wires with different sequences because they can be recognized on detailed comparisons of their UV-vis and fluorescence spectra (Figure 5d,e).
- (b) It is well-known that the absorption peak and redox peak of complex can be altered by modifying or changing organic ligands. As an alternative method for sequence decoding in future, the sequence decoding will become easy if 10 kinds of complexes containing the identical metal core and different organic ligands with their own specific absorption and redox features are used for iterative synthesis at specific positions in single organometallic wire.

We have added this answer into paper.

c. In general, the authors make broad generalizations that these materials will provide “significant advantages of appealing information-related applications...” – more context as to the current state of the field with different (bio)polymer strategies would be valuable.

Response 15:

We have modified and added new sentences before Figure 5 as following:

Generally, every monomer in digital macromolecule expresses 0- or 1- bit, and macromolecule with 8 monomers gives a letter in ASCII (American Standard Code for Information Interchange).¹⁷ In this paper, every monomer (metal core) can express one or two (c.a. Ru, Os, Fe) letter, and single organometallic wire could express a word or sentence because of a number of available ligands and metal species with rich-stimuli-responsivestates. 10 and 20 kinds of letter in single organometallic wire containing 10 monomers have the sequences of over 3.6×10^6 and 6.7×10^{11} . Therefore, these organometallic wires theoretically have expectably ultrahigh information storage with exponential enhancement compared with single kind of metal cores.¹⁸ A large library of sequences and hard characterization ensure these organometallic wires to have ultrahigh anti-counterfeiting security. Additionally, the electrochemistry and UV-vis spectra are considered to be the convenient and low-cost coding and decoding processes for sample in both liquid and solid states compared to other techniques such as NMR and Mass spectra.

2. The comparison to solid-phase synthesis, and the inefficiency of using an excess of reagents that are washed away, is made multiple times. However, this synthetic strategy similarly exchanges the solution with every monomer addition. Are these solutions reused? Are lower concentrations of reagent required than solid-phase synthesis on a comparable surface?

Response 16:

Two solutions of identical monomer were respectively reused for oxidative coupling and reductive coupling. In this case, the dimer obtained from monomer self-coupling in solution does not further react with molecules on substrate.

In this study, the concentration of monomers is 0.5 mM. Theoretically, this method does not need high concentration as well as solid-phase synthesis because of electric stimuli.

3. Minor comments:

- a. Figure 1 is hard to follow – can this be edited to be easier to follow for an external audience?

Additionally, the aromatic ring structures have different lengths for the double bonds.

Response 17:

Similar to Response 9, we have redesigned Figure 1 in multistep reactions for easy understanding. There are the different lengths in aromatic ring because of hetero-atomic structure. We have redrawn the molecular structures.

b. For the plots where the x-axis is labeled “switching time” – should this have a unit affiliated?

Response 18:

We have modified all figures in manuscript and supporting information, and “switching time” is replaced by “switching times (n)”.

c. Some of the graphs appear to have a linear regression fit – can the R^2 value and any additional relevant information be provided?

Response 19:

We have added R^2 value in Figure 2 and Figure S5. The value of R^2 is closer to 1, indicating better linear relationship of data for uniform synthesis.

Reviewers' comments:

Reviewer #1 (Remarks to the Author):

The additional work that the authors have put into this communication is substantial and commendable. It's clear the effort that was undertaken to compile a great deal more experimental data on the characterization of these "organometallic wires." Unfortunately, these efforts seem to have further put into question the claims of this communication.

Regardless of if the compounds in question are called "polymers," "macromolecules," or "organometallic wires," the term sequence-defined has a very precise meaning. For a compound to be sequence-defined, it needs to be of EXACT chain length and have a PERFECTLY defined sequence of monomers (Science, 341,1238149). I do not believe the compounds in this communication cannot be considered "sequence-defined" for 3 major reasons.

1) The first concern comes from the organometallic bonds, which for the most part are known to be reversible/dynamic. While asymmetric bis(terpy) complexes of Os(III), Rh(III), Ru(III), Ir(III) are known to be relatively stable, other complexes of terpy with metals like Fe(II) and Co(II) are, to my understanding, not as stable, and readily depolymerize and exchange ligands (Macromol. Rapid Commun. 2010, 31, 784). The additional data provided in the authors' rebuttal does not support that there is no ligand exchange between the synthesized "organometallic wires," especially those containing Fe units. The effort into the additional experiments the authors have performed is appreciated: characterization of the dimer is especially interesting. I appreciate that these compounds are very difficult to characterize given their solubility and instability. At the same time, there isn't evidence to demonstrate that there aren't any ligand exchanges occurring over time that would degrade the original monomer sequence. Other individuals have made very similar chemical structures to those described in this communication using similar processes of stepwise chemistry for layer-by-layer assemblies of organometallic wires and have not made similarly unfounded claims of "sequence-definition" (Coord. Chem. Rev. 346, 139-149 (2017); J. Phys. Chem. C 2018, 122, 3419–3427).

2) The second concern comes from the inability to adequately show quantitative conversion after each coupling reaction, something that all solid-phase and other sequenced-defined synthesis needs to demonstrate to be credible. The authors simply cannot compare their methodology to solid-phase synthesis or other iterative methods. As is required for validating other methodologies, the authors have not demonstrated that their coupling reactions are efficient enough to proceed quantitatively. For example, the authors suggest the organic wire in Figure 3a supposedly has an average sequence of 1111122222 (1 = Os, 2 = Fe). We do not and cannot know how pure these chains are due to instability and insolubility. It is very likely to be contaminated with a large dispersity of shorter sequences like 111122222, 111122, 11222, etc. due to incomplete reactivity at each coupling reaction. UV-abs and CV measurements are just not enough to make a claim of quantitative addition. Given the insolubility of the chains in any solvent and consequently, dubious accessibility to the end groups of the growing "organometallic wires," it is heavily doubtful that the compounds in this communication aren't contaminated with shorter chains (Solid-phase synthesis, for example, can only work in certain solvents that sufficiently swell the solid support and expose the growing polymer chain ends to the coupling reagents). This dispersity would hardly allow these organometallic wires to qualify as "sequence-defined," aka having "EXACT chain length." Solid-phase synthesis had to undergo decades of optimization (coupling reaction reagents, solvents to swell the solid support, efficient cleavage reactions from the solid support, etc.) before it could become a versatile method to make sequence-defined polymers. It's understandable that the compounds in this communication cannot be thoroughly characterized, but then the authors cannot claim "sequence-defined compounds." You cannot claim that which you cannot provide sufficient proof for.

3) There's no presented evidence that the carbazole coupling isn't forming regioisomers other than

the C3-C3' bond with each oxidative coupling reaction. The unclear ¹H NMR of the dimer in Figure S18 puts the efficiency of the carbazole coupling reaction completely into question. Figure S18 seems to indicate the carbazole coupling is not completely regioselective and may indeed lead to regioisomers as evidenced by the significant impurity/shoulder peaks at 9.18, 8.50-8.56, and 8.28-8.33 ppm. These peaks are not minor and indicate at least 10% of some kind of contaminant. If we're generous, and assume there's even a 95:5 ratio of C3-C3' to other undesired regioisomer couplings, that would mean after the 10th coupling (assuming quantitative conversion, which has not been demonstrated) only 60% (0.95^{10}) of the grown chains are the predicted, desired structure.

It may also be possible that the extraneous peaks in the ¹H NMR of the dimer are just from unreacted Ru XY starting material, but if that's the case, it just bolsters the second major concern that this electrochemical oxidative coupling reaction is not quantitative, and there will be unavoidable impurities from unreacted chains.

Chemistry to make "sequence-defined" polymers has advanced greatly in the past few decades and in 2020, needs to be held to a greater standard. The data and lack thereof in the communication indicates the "organometallic wires" in this communication are disperse and by definition, not "sequence-defined." Based on these reasons, I cannot recommend the communication be accepted as is. The communication claims more than what it can prove.

Additionally, can these compounds be called "organometallic wires" when they're not fully conjugated and consequently conductive? The aliphatic region of each unit prevents electric current from traveling through the "wire." Other organometallic wires from my understanding can all conduct electric current (Coord. Chem. Rev. 346, 139-149 (2017)).

Additionally, it is somewhat difficult to justify the novelty and importance of the synthetic methodologies presented in this communication when there are more elegant, efficient examples in publications from more specialized journals.

1) [Yamanoi, Y. JACS, 2012, 134, 20433] is an early work that demonstrates the synthesis of well-defined 10-layer Fe molecular wires through sequential coordination chemistry of Fe and bis(terpyridine) monomers. They, like the authors of the communication, found that with each additional layer, the overall properties of their molecular wires significantly changed in their CVs, STMs, current-time plots, etc. The chemical structures are very similar to those shown in the submitted manuscript.

2) [Nishimori, Y. Chem. Asian J. 2007,2, 367-376] demonstrates the layer-by-layer synthesis of well-defined dendritic Fe molecular wires with up to 15 metal complexes and well-defined linear Fe molecular wires up to 8 units long. They, like the authors of this communication, demonstrate that varying the lengths of these wires significantly change the CVs, AFMs, current-time plots, etc. The structures are similar to the structures described in the submitted manuscript, except instead of using a dicarbazolyl unit in between the terpyridines, Nishimori et al. use diazo or alkyne functionalities.

3) [Poisson, J. J. Phys. Chem. C., 2018, 122, 3419.] synthesized Fe molecular wires with either linear and zigzag architectures with 1 to 12 layers. Each additional layer, like in previous examples, affords significantly different properties. Again, structurally very similar to the structures in the submitted manuscript.

There are many other examples in the literature that build organometallic wires through step-by-step, layer-by-layer chemistry, just like what this communication presents. To justify the significance and novelty of their chemistry, the authors need to provide a compelling argument for how their synthesis is superior to past methods. At this point, it seems like step-by-step metal coordination that prior work uses could easily make the compounds presented in this communication in a more efficient and simple way.

Still, the data (Figures 3-5) that demonstrates that changing the sequence of monomer additions with different metals leads to different physical properties is certainly novel and of interest. The paper cannot call these functionalized surfaces "sequence-defined," but clearly the sequence of the chemistry does matter. It's still questionable whether these different properties can be applied to information storage, but again the overall proof that "sequence matters" is novel.

This communication needs to be rewritten in a major way that deemphasizes claims of "sequence-definition" that it cannot substantiate. It also needs to justify the novelty of the chemistry or also deemphasize its importance in the communication. I would recommend the communication's introduction focus more heavily on the novelty of using different sequential addition a variety of metals to make novel materials for layer-by-layer surface modification.

Other notes:

Figure 1 is still immensely confusing and far from publication quality. Solid/dotted arrows and the directions they point to do not seem to have consistent meaning. Double bonds are inconsistent not only in the aromatic groups, but also in the monosubstituted alkenes. The structures are bizarrely slanted. If I were to suggest how to improve it, please draw all structures following recommendations by the ACS Style Guide. Draw out full chemical representations of the final structures. Please refer to figures in [Coord. Chem. Rev. 346, 139-149 (2017)] which, in my opinion, are exceptionally done.

References are not in Nature Communications format. Only use numerical references for citations. Do not use "a)...b)...c)...etc." to differentiate references.

Supporting information issues:

- 1) It is not acceptable to not have a ^{13}C NMR of each small molecule building block and metal complex monomer.
- 2) Yields need to be provided for every reaction (Ru XP, Os XP, Os X2).
- 3) Full reaction procedures and reagent quantities need to be provided for all reactions. Simply saying the "target molecule was obtained by using the similar protocol of synthesis..." is not acceptable for a synthetic paper.
- 4) Significant figures are all over the place and need to be edited to be consistent.
- 5) Check for typos. For example, line 371, it should be $(\text{NH}_4)_2\text{OsCl}_6$, not " $(\text{NH}_4)\text{OsCl}_6$ "
- 6) Figure S33 and S34 should have cleaner NMRs. These compounds are too simple to have such visible impurities.
- 7) ALL small molecules need at minimum LRMS characterization, if not HRMS. There's no reason to not have this simple characterization.

Carefully check grammar in the Supporting Information. For example...

Line 71, "appears" should be "appear"

Line 72, "there is" should be "there are"

Line 73, "within [the] self-assembled monolayer"

Line 74, the CVs confirm the completion of the self-coupling reactions of the carbazoyls. It doesn't make sense for reactions to be found in CVs.

Line 83, "band[s]"

Line 85, "[Moderate] absorption bands [in] the range of 490-600 nm"

...Etc.

Reviewer #2 (Remarks to the Author):

The authors have made significant changes, and answer my questions. Therefore i confirm my previous recommendation: publication.

Reviewer #3 (Remarks to the Author):

The authors have addressed all of my concerns, I recommend this for publication.

We do understand all comments. We highly appreciate the patient reviewer's comments, which lead our paper to be more readable and also have filled us with awe and respect. The reviewer's concerns were partially addressed into manuscript in visible revision mode and supporting information in response mode in this letter.

Point-by-point response to the reviewer comments:

Reviewers' comments:

Reviewer #1 (Remarks to the Author):

The additional work that the authors have put into this communication is substantial and commendable. It's clear the effort that was undertaken to compile a great deal more experimental data on the characterization of these "organometallic wires." Unfortunately, these efforts seem to have further put into question the claims of this communication.

Regardless of if the compounds in question are called "polymers," "macromolecules," or "organometallic wires," the term sequence-defined has a very precise meaning. For a compound to be sequence-defined, it needs to be of EXACT chain length and have a PERFECTLY defined sequence of monomers (Science, 341,1238149). I do not believe the compounds in this communication cannot be considered "sequence-defined" for 3 major reasons.

Response 1:

The "sequence-defined" is replaced by "sequence-controlled".

The "organometallic wire" is replaced by "organometallic polymer" in this manuscript. Organometallic polymer is well-used for conjugated and non-conjugated polymers:

Angew. Chem. Int. Ed. 1999, 38, 2570; *Adv. Mater.* 2003, 15, 51; *Angew. Chem. Int. Ed.* 2005, 44, 2568; *Macromolecules* 2006, 39, 3786; *Angew. Chem. Int. Ed.* 2007, 46, 9069; *J. Am. Chem. Soc.* 2009, 131, 5378; *J. Am. Chem. Soc.* 2014, 136, 7865; *ACS Macro Lett.* 2015, 4, 593; *Macromolecules* 2018, 51, 1351.

1) The first concern comes from the organometallic bonds, which for the most part are known to be reversible/dynamic. While asymmetric bis(terpy) complexes of Os(III), Rh(III), Ru(III), Ir(III) are known to be relatively stable, other complexes of terpy with metals like Fe(II) and Co(II) are, to my understanding, not as stable, and readily depolymerize and exchange ligands (Macromol. Rapid Commun. 2010, 31, 784).

Response 2:

This reference is added as new Ref. 37 and S6.

We agree with this concern. We do think that this concern brings us an interesting research topic for future research. However, this study was performed with bare tpy in solution in most cases. The organometallic wires in solid-like state are essentially different. For our system without bear tpy, these behaviors could be greatly restricted, within the organometallic molecular wires closely packed

and assembled on electrode, and between molecules assembled on electrode and molecule in solution. Actually, we did not find any experiment phenomenon, which could lead us to think about the possibility for depolymerization and ligand exchange.

In this concern, the synthesis and purification procedures for each addition took for very long time up to 24h for stepwise chemical coordination. Our electrocatalysis with each addition of monomer of 1 min will be favorable to achieve highly structural controllability.

The additional data provided in the authors' rebuttal does not support that there is no ligand exchange between the synthesized "organometallic wires," especially those containing Fe units. The effort into the additional experiments the authors have performed is appreciated: characterization of the dimer is especially interesting. I appreciate that these compounds are very difficult to characterize given their solubility and instability. At the same time, there isn't evidence to demonstrate that there aren't any ligand exchanges occurring over time that would degrade the original monomer sequence. Other individuals have made very similar chemical structures to those described in this communication using similar processes of stepwise chemistry for layer-by-layer assemblies of organometallic wires and have not made similarly unfounded claims of "sequence-definition" (Coord. Chem. Rev. 346, 139-149 (2017); J. Phys. Chem. C 2018, 122, 3419–3427).

Response 3:

Their groups do not have the background of conventional polymer synthesis. We think that they did not claim probably because their consideration was not subjective to contribute their work to polymer synthesis.

Our electrocatalysis has good controllability to enable the **rapid and uniform** synthesis (**1 min for each addition**) of organometallic polymers, compared to well-known method based on chemical coordination. Usually, the time-consuming stepwise (**c.a. overnight in "Chem. Commun., 2013, 49, 7108." or 24 h in "J. Phys. Chem. C 2018, 122, 3419." for each addition**) incorporation of metal cores in previous reports did not have good linear relationships between units and steps. It is unable for the uniform synthesis of organometallic wires, and will be difficult for further synthesis of sequence controlled organometallic polymers.

Selected references for last 10 years as Ref. 27-29 were added into manuscript for appealing discussion because the experiments updated for last 10 years should be more reliable for researchers.

Example 1: *Nat. Mater.* 8, 41-46 (2009).

Figure 2 | Characterization of films for Fe(II)-based MCMWs by ToF-SIMS and ultraviolet-visible spectroscopy. **a,b**, Typical ToF-SIMS spectra for samples after one and five coordination steps respectively. The intensity of the peaks characteristic of the platform monolayer ($[\text{C}_{21}\text{H}_{14}\text{N}_3\text{SAu}]^+$) at 536 Da as well as those characteristic of the Au substrate (around 540 Da) decreases and the intensity of complex-related peaks (for example $[\text{C}_{36}\text{H}_{24}\text{N}_6\text{Fe}]^+$) at 596 Da increases, as expected from the fact that the technique is more sensitive to the uppermost layers. **c**, Shift of the surface-plasmon maximum energy of the gold substrate as a function of the number of coordination steps, n ($n \leq 6$). **d**, Optical density at the maximum of the MLCT band ($\lambda_{\text{max}} = 598 \text{ nm}$) as a function of the number of coordination steps ($n > 6$).

Though the addition of each monomer costs 15 min in this paper, there is no full data of all additions.

Example 2: *Coord. Chem. Rev.* 346, 139-149 (2017)

Example 3: *J. Phys. Chem. C* 122, 3419–3427 (2018)

Figure 1. Terpyridine based molecular blocks 1 and 2 and schematic representation of the molecular assemblies 1-Fe and 2-Fe on the monoterpy template layer (TL). Charges on the metal center and counteranions are omitted for clarity.

Figure 4. Optical absorption spectra of (A) 1-Fe and (B) 2-Fe after each deposition cycle. Electrochemical tests and DP voltammograms for (C) 1-Fe and (D) 2-Fe. (E) Intensity of MLCT absorption band (587 nm for 1-Fe and 568 nm for 2-Fe assembly) as a function of deposition cycles. (F) Charge density dependence on the number of deposition cycles. (G) Variation in surface coverage in mols of iron cm^{-2} as a function of the number of deposition cycles. (H) Relationship between MLCT band intensity and charge density for both assemblies.

Figure 4A-D did not show good regular increases in absorbance and current intensities. For monomer 1-Fe in Figure 4E-H, 1st-4th additions and 5-11th additions are not in linear relationship. For monomer 2-Fe in Figure 4E-H, the linear relationships (R^2 down to 0.95) should have large error

because of obvious shifts of peak positions (Figure 4A-D), while there is no complete data for each addition.

2) *The second concern comes from the inability to adequately show quantitative conversion after each coupling reaction, something that all solid-phase and other sequenced-defined synthesis needs to demonstrate to be credible. The authors simply cannot compare their methodology to solid-phase synthesis or other iterative methods. As is required for validating other methodologies, the authors have not demonstrated that their coupling reactions are efficient enough to proceed quantitatively. For example, the authors suggest the organic wire in Figure 3a supposedly has an average sequence of 1111122222 (1 = Os, 2 = Fe). We do not and cannot know how pure these chains are due to instability and insolubility. It is very likely to be contaminated with a large dispersity of shorter sequences like 111122222, 111122, 11222, etc. due to incomplete reactivity at each coupling reaction. UV-abs and CV measurements are just not enough to make a claim of quantitative addition. Given the insolubility of the chains in any solvent and consequently, dubious accessibility to the end groups of the growing “organometallic wires,” it is heavily doubtful that the compounds in this communication aren’t contaminated with shorter chains (Solid-phase synthesis, for example, can only work in certain solvents that sufficiently swell the solid support and expose the growing polymer chain ends to the coupling reagents). This dispersity would hardly allow these organometallic wires to qualify as “sequence-defined,” aka having “EXACT chain length.” Solid-phase synthesis had to undergo decades of optimization (coupling reaction reagents, solvents to swell the solid support, efficient cleavage reactions from the solid support, etc.) before it could become a versatile method to make sequence-defined polymers. It’s understandable that the compounds in this communication cannot be thoroughly characterized, but then the authors cannot claim “sequence-defined compounds.” You cannot claim that which you cannot provide sufficient proof for.*

Response 4:

In our paper, the impurities (c.a. shorter chains) cannot be denied. For almost organic synthesis, the impurities are absolutely unavoidable. It is well-known that solution-process synthesized soluble polymers allow second and more purification processes for significant purity with good structural characterizations. However, all scientists probably also would like to pay attentions on the purity availability for further applications.

Regarding the purity, the purity of organometallic polymers is considered to depend on the monomer size, the reaction time and frequency, and the electrochemical condition (trace O₂ and water, roughness of electrode). In particular, the unit size of ITO substrate has the surface coverage (Γ) of 10⁻¹⁰ mol cm⁻² in Figure S4, which could not support more space for more large size monomers with same surface coverage for next step. Therefore, the size changes of different monomers are not favorable for linear relationship of units and steps. All monomers in our paper are considered to have the same size.

In our paper, the Os surface coverage of resulting organometallic polymer has excellent linear relationships (up to 0.998 in Figure 2 and S5, probably best among all papers reported), indicating

the dispersity of polymers could be ignored for further applications. Thus, the single monomer addition and the length of organometallic polymers can possibly tend to quantitative and uniform synthesis. For our current electrochemical condition, the electrochemical cell is open to air with sample argon bubble. We believe that the purity of organometallic polymers can be further optimized by the reaction time and times for each step.

Regarding the purity availability for further applications, the data of micro or nano-sized units of substrates will be evaluated. It is important that each addition of the units to organometallic polymers should be recognized, while the signal of impurity can be ignored. In our paper, each addition of all units to organometallic polymers can be well-recognized in figures of UV-vis spectra and CV, demonstrating an applicable potential.

3) *There's no presented evidence that the carbazole coupling isn't forming regioisomers other than the C3-C3' bond with each oxidative coupling reaction. The unclear 1H NMR of the dimer in Figure S18 puts the efficiency of the carbazole coupling reaction completely into question. Figure S18 seems to indicate the carbazole coupling is not completely regioselective and may indeed lead to regioisomers as evidenced by the significant impurity/shoulder peaks at 9.18, 8.50-8.56, and 8.28-8.33 ppm. These peaks are not minor and indicate at least 10% of some kind of contaminant. If we're generous, and assume there's even a 95:5 ratio of C3-C3' to other undesired regioisomer couplings, that would mean after the 10th coupling (assuming quantitative conversion, which has not been demonstrated) only 60% (0.95^{10}) of the grown chains are the predicted, desired structure. It may also be possible that the extraneous peaks in the 1H NMR of the dimer are just from unreacted Ru XY starting material, but if that's the case, it just bolsters the second major concern that this electrochemical oxidative coupling reaction is not quantitative, and there will be unavoidable impurities from unreacted chains.*

Response 5:

We do understand this comment. Now, these comments are addressed into Figure S17-18. We have tried to get the isolated products from excess supporting electrolytes mostly via silica gel chromatography. This experiment failed because of the dimer with counterion is eluted out along with $\text{Bu}_4\text{NH}_4\text{ClO}_4$.

Figure S17. Photos of $\text{Ru}^{\text{II}}\text{XY}$ solutions before (a) and after (b) iterative synthesis, and photo of bare ITO electrode without pre-assembled molecule after electrolysis of $\text{Ru}^{\text{II}}\text{XY}$ at 1.0 V for 1h (c). The precipitation can be found after iterative synthesis, indicating the oligomers become insoluble, and also implying that the single organometallic polymer with 10 units will be hard to be soluble. After

electrolysis of Ru^{II}XY at 1.0 V for 1h, the dark red film was found on ITO surface (c) and directly dissolved without any purification for measurements of NMR and Mass spectra as shown in Figure S18,19, demonstrating that this film was mainly composed of dimer with bad solubility. We have tried to get the isolated products from excess supporting electrolytes mostly via silica gel chromatography. This experiment failed because of the dimer with counterion is eluted out along with Bu₄NH₄ClO₄.

Figure S18. ¹H NMR spectra of Ru^{II}XY and its dimer in CD₃CN obtained by electrolysis in Figure 17c without any purification. Regarding the impurity feature in Figure S18b, the unreactive monomer could possibly and physically co-deposit into film during continuously electrolysis of 1h. We did not find the clear evidence that the carbazole coupling is forming regioisomers for these monomers or other monomers, which we were and are studying on. Herein, this possibility should be limited in iterative electrosynthesis because of steric hindrance. Additionally, it is well-known that the purity will decrease in case of large scale synthesis. Compared to synthesis on 1 cm² substrate (10⁻¹⁰ mol and 10⁻⁷ g for each step in Figure S4), the experiment in Figure S18 (c.a. 2 mg) was enlarged in over 10000 times. Iterative synthesis in manuscript took place at a distance of 20 nm from electrode surface, while this electrodeposition fabricated the film with probable thickness of >10 μm. Therefore, the coupling ratio in Figure S18 and the coupling ratio of each step for iterative synthesis of organometallic polymer could not be simply compared. The relationship should be a clear curve even if there was significant decrease in total conversion (c.a. 10% after 10th coupling). The relationship between units and steps in Figure 2 shows excellent linear, demonstrating that there is no significant change in conversion yield.

Regarding the possible regioisomers other than the C3-C3' bond:

With controlled oxidation strength, C3-C6 and C3-C3' couplings give same structure of dimer in solution, but they will give different structures if there were molecules assembled on electrode surfaces.

For organometallic polymers, the regioisomers could exist.

Chemistry to make “sequence-defined” polymers has advanced greatly in the past few decades and in 2020, needs to be held to a greater standard. The data and lack thereof in the communication indicates the “organometallic wires” in this communication are disperse and by definition, not “sequence-defined.” Based on these reasons, I cannot recommend the communication be accepted as is. The communication claims more than what it can prove.

Response 6:

“Sequence-defined” was replaced by “Sequence-controlled” in this manuscript.

Additionally, can these compounds be called “organometallic wires” when they’re not fully conjugated and consequently conductive? The aliphatic region of each unit prevents electric current

from traveling through the “wire.” Other organometallic wires from my understanding can all conduct electric current (*Coord. Chem. Rev.* 346, 139-149 (2017)).

Response 7:

We have checked this reference and other papers, and found that there is almost no “organometallic wire” used in paper title. The molecular wire was most used, and probably emphasized with additionally desirable words (complex, conductive, etc.).

We have found the “organometallic polymer” is accepted. The organometallic polymer can be conjugated or non-conjugated. Therefore, the “wire” is replaced by “polymer” in manuscript.

Additionally, it is somewhat difficult to justify the novelty and importance of the synthetic methodologies presented in this communication when there are more elegant, efficient examples in publications from more specialized journals.

Response 8:

Similarly to Response 2:

We have added these sentences into manuscript:

Electrosynthesis is **rapid (1 min for each addition)**, and **independent on metal species**, and it provides **significantly high controllability toward uniform synthesis**, compared to well-known method based on chemical coordination. To date, the organometallic polymers were synthesized mostly by the iterative metal coordination between **almost Fe²⁺** and **tpy** ligands on solid substrate, which take **10~24 h** for each addition of single monomer at room temperature.²⁷⁻²⁹

Usual chemical coordination for synthesis of organometallic wires²⁷⁻²⁹ requires the high-quality solvents without external ions for synthesis and purification, and its controllability and reproducibility still remain stagnant and challenge in general metal species and ligand species, and following coordination types for further sequence controlled synthesis.

Selected 3 papers in last 10 years as Ref 27-29:

Yamanoi, Y., Sendo, J., Kobayashi, T., Maeda, H., Yabusaki, Y., Miyachi, M., Sakamoto, R. & Nishihara, H. A new method to generate arene-terminated Si(111) and Ge(111) surfaces via a palladium-catalyzed arylation reaction. *J. Am. Chem. Soc.* 134, 20433-20439 (2012).

Sakamoto, R., Ohirabaru, Y., Matsuoka, R., Maeda, H., Katagiri, S. & Nishihara, H. Orthogonal bis(terpyridine)-Fe(II) metal complexoligomer wires on a tripodal scaffold: rapid electrontransport. *Chem. Commun.* 49, 7108-7110 (2013).

Poisson, J., Geoffrey, H. L., Ebralidze, I. I., Laschuk, N. O., Allan, J. T. S., Deckert, A., Easton, E. B. & Zenkina, O. V. Layer-by-layer assemblies of coordinative surface-confined electroactive multilayers: zigzag vs orthogonal molecular wires with linear vs molecular sponge type of growth. *J. Phys. Chem. C* 122, 3419-3427 (2018).

1) [Yamanoi, Y. *JACS*, 2012, 134, 20433] is an early work that demonstrates the synthesis of well-defined 10-layer Fe molecular wires through sequential coordination chemistry of Fe and bis(terpyridine) monomers. They, like the authors of the communication, found that with each additional layer, the overall properties of their molecular wires significantly changed in their CVs,

STMs, current-time plots, etc. The chemical structures are very similar to those shown in the submitted manuscript.

Response 8:

The reaction time of each addition is 23h at room temperature.

Reprinted (adapted) with permission from J. Am. Chem. Soc. 2012, 134, 50, 20433-20439. Copyright (2012) American Chemical Society.

Figure 5. Preparation of Si(111)-4 and resulting oligomer wires. (i) Immobilization of 4'-(4-iodophenyl)-2,2':6',2''-terpyridine (**4**) on Si(111). Si(111)-H was immersed in a 1,4-dioxane solution (7.5 mL) of Pd(P(*t*-Bu)₃)₂ (0.3 mg), (*i*-Pr)₂EtN (0.38 mL), and **4** (0.06 mmol) at 100 °C for 3 h. (ii, iii) Preparation of Fe(tpy)₂ complex oligomer wires, Si(111)-4-(FeL)_n. Inset: Cyclic voltammograms for Si(111)-4-(FeL)_n (*n* = 1-4) taken at a scan rate of 0.1 V s⁻¹ in 1 M Bu₄NClO₄/CH₂Cl₂ (left) and coverage of Si(111)-4-(FeL)_n versus the number of coordination cycles (right).

In this figure, there is lack of full data for every addition, while the current increase looks not regular and the data dots are offline.

Figure 6. (a) Representative AFM images of Si(111)-4, Si(111)-4-(FeL)₂, Si(111)-4-(FeL)₄, and Si(111)-4-(FeL)₁₀. (b) Histogram of height analyses of Si(111)-4, Si(111)-4-(FeL)₂, Si(111)-4-(FeL)₄, and Si(111)-4-(FeL)₁₀.

This figure indicates that the molecule does not fully cover the substrate, and shows the clear distribution of molecular length. Most of papers published previously remain challenge in acceptably uniform synthesis. In Figure S8 of our paper, there is only ~2 nm change in heights. In Figure 2, the

linear relationship demonstrates the excellent controllability, which enables superior iterative synthesis of multiple monomers in our paper.

2) [Nishimori, Y. *Chem. Asian J.* 2007,2, 367–376] demonstrates the layer-by-layer synthesis of well-defined dendritic Fe molecular wires with up to 15 metal complexes and well-defined linear Fe molecular wires up to 8 units long. They, like the authors of this communication, demonstrate that varying the lengths of these wires significantly change the CVs, AFMs, current-time plots, etc. The structures are similar to the structures described in the submitted manuscript, except instead of using a dicarbazolyl unit in between the terpyridines, Nishimori et al. use diazo or alkyne functionalities.

Response 9:

The reaction time of each addition is 3h at room temperature. We did not cite this paper because their group took the reaction time of 23 h in 2012. (*J. Am. Chem. Soc.* 134, 20433-20439 (2012).)

Figure 3. Plots of the coverage of redox-active sites, Γ , versus the number of complexation cycles, n , for A) $[n\text{Fe}3]$ ($t_1=5$ min), B) $[n\text{Fe}4]$ ($t_1=5$ min), and C) $[1\text{Fe}3-(n-1)\text{Fe}4]$ ($t_1=10$ s). The lines denote the relationships $\Gamma=C \times n$ for A, $\Gamma=C(2^n-1)$ for B, and $\Gamma=C2^{n-1}$ ($n \geq 1$) for C.

Linear relationship is not good. They did not show the absorption spectra.

3) [Poisson, J. J. *Phys. Chem. C.*, 2018, 122, 3419.] synthesized Fe molecular wires with either linear and zigzag architectures with 1 to 12 layers. Each additional layer, like in previous examples, affords significantly different properties. Again, structurally very similar to the structures in the submitted manuscript.

Response 10:

The reaction time of each addition is 24h at room temperature.

Reprinted (adapted) with permission from (J. Phys. Chem. C 2018, 122, 6, 3419-3427). Copyright (2018) American Chemical Society.

Figure 1. Terpyridine based molecular blocks 1 and 2 and schematic representation of the molecular assemblies 1-Fe and 2-Fe on the monoterpy template layer (TL). Charges on the metal center and counteranions are omitted for clarity.

Figure 4. Optical absorption spectra of (A) 1-Fe and (B) 2-Fe after each deposition cycle. Electrochemical tests and DP voltammograms for (C) 1-Fe and (D) 2-Fe. (E) Intensity of MLCT absorption band (587 nm for 1-Fe and 568 nm for 2-Fe assembly) as a function of deposition cycles. (F) Charge density dependence on the number of deposition cycles. (G) Variation in surface coverage in mols of iron cm^{-2} as a function of the number of deposition cycles. (H) Relationship between MLCT band intensity and charge density for both assemblies.

Figure 4A-D did not show good regular increases in absorbance and current intensities. For monomer 1-Fe in Figure 4E-H, 1st-4th additions and 5-11th additions are not in linear relationship. For monomer 2-Fe in Figure 4E-H, the linear relationships (R^2 down to 0.95) should have large error

because of obvious shifts of peak positions (Figure 4A-D), while there is no full data for each addition.

There are many other examples in the literature that build organometallic wires through step-by-step, layer-by-layer chemistry, just like what this communication presents. To justify the significance and novelty of their chemistry, the authors need to provide a compelling argument for how their synthesis is superior to past methods. At this point, it seems like step-by-step metal coordination that prior work uses could easily make the compounds presented in this communication in a more efficient and simple way.

Response 11:

With our best efforts during last 3 months, most of paper available on website have been downloaded most of papers, which could be found on website. All papers reported previously were studied based on almost only Fe²⁺, their availability for other metal species remains questions (for example, C-metal types of complexes). **Here electroynthesis not only shows the rapid synthesis and general potential for a lot of types of complexes with different metal species and organic ligands, but also provides a highly reliable controllability in sequence controlled synthesis of organometallic polymers.**

Similarly to Response 8:

We have added these sentences into manuscript:

Electrosynthesis is **rapid (1 min for each addition)**, and **independent on metal species**, and it provides **significantly high controllability toward uniform synthesis**, compared to well-known method based on chemical coordination. To date, the organometallic polymers were synthesized mostly by the iterative metal coordination between **almost Fe²⁺** and **tpy** ligands on solid substrate, which take **10~24 h** for each addition of single monomer at room temperature.²⁷⁻²⁹

Usual chemical coordination for synthesis of organometallic wires²⁷⁻²⁹ requires the high-quality solvents without external ions for synthesis and purification, and its controllability and reproducibility still remain stagnant and challenge in general metal species and ligand species, and following coordination types for further sequence controlled synthesis.

Here electroynthesis not only shows the rapid synthesis and general potential for a lot of types of complexes with different metal species and organic ligands, but also provides a highly reliable controllability in sequence controlled synthesis of organometallic polymers.

Still, the data (Figures 3-5) that demonstrates that changing the sequence of monomer additions with different metals leads to different physical properties is certainly novel and of interest. The paper cannot call these functionalized surfaces “sequence-defined,” but clearly the sequence of the chemistry does matter. It’s still questionable whether these different properties can be applied to information storage, but again the overall proof that “sequence matters” is novel.

Response 12:

The small molecules and polymers with electrochemical and optical responses were suggested for information storage by worldwide researchers. Usually, the electrochemical and optical responses are very simple. (J. Phys. Chem. C, 2009, 113, 8548-8552. Angew. Chem. Int. Ed. 2006, 45, 2016-2035.

Adv. Mater. 2005, 17, 459-464. Adv. Mater. 2005, 17, 156-160. J. Am. Chem. Soc. 2012, 134, 20053-20059.)

By utilizing our method, single organometallic polymer can provide the rich electrochemical in wide potential range (c.a. -2.5 V to 2 V, depends on stability of complexes) and optical responses as more as we can add the distinguishable complexes.

Regarding digital polymers for information storage:

Acc. Chem. Res. 2013, 46, 2696-2705. Nat. Chem. 2019, 11, 136-145. Macromolecules 2015, 48, 4759-4767. Polymer 1997, 38, 3767-3781.

This communication needs to be rewritten in a major way that deemphasizes claims of "sequence-definition" that it cannot substantiate. It also needs to justify the novelty of the chemistry or also deemphasize its importance in the communication. I would recommend the communication's introduction focus more heavily on the novelty of using different sequential addition a variety of metals to make novel materials for layer-by-layer surface modification.

Response 12:

This response is similar to Response 1, 3, 8, 11.

Other notes:

Figure 1 is still immensely confusing and far from publication quality. Solid/dotted arrows and the directions they point to do not seem to have consistent meaning. Double bonds are inconsistent not only in the aromatic groups, but also in the monosubstituted alkenes. The structures are bizarrely slanted. If I were to suggest how to improve it, please draw all structures following recommendations by the ACS Style Guide. Draw out full chemical representations of the final structures. Please refer to figures in [Coord. Chem. Rev. 346, 139-149 (2017)] which, in my opinion, are exceptionally done.

Response 13:

We have redrawn the scheme and molecular structures according ACS and paper [Coord. Chem. Rev. 346, 139-149 (2017)] styles.

References are not in Nature Communications format. Only use numerical references for citations. Do not use “a)...b)...c)...etc.” to differentiate references.

Response 15:

We have modified this part.

Supporting information issues:

1) It is not acceptable to not have a ^{13}C NMR of each small molecule building block and metal complex monomer.

Response 16:

We have added ^{13}C NMR spectra for ligands (new Figure S34, S37, S40), and for $\text{Ru}^{\text{II}}\text{XY}$ and $\text{Os}^{\text{II}}\text{XY}$ monomers (new Figure S44 and S47). The monomers have low solubility in MeCN as best solvent for NMR experiments, and the 0.4~1.5 mg/sample were taken for 5~7 h. We did not get good ^{13}C NMR spectra of $\text{Ir}^{\text{III}}\text{XY}$, $\text{Ru}^{\text{II}}\text{XP}$, and $\text{Os}^{\text{II}}\text{XP}$ (new Figure S50, S56, S59) because of very low solubility of these monomers.

^{13}C NMR spectrum of $\text{Os}^{\text{II}}\text{X}_2$ is not provided at this time because we do not have this compound now and the students are waiting home for unknown end of coronavirus. ^{13}C NMR spectrum of $\text{Os}^{\text{II}}\text{X}_2$ is not key data for our description in manuscript, and its purity does not affect our results. We would like to present it in future if data availability was needed in case of curious readers because we mention that all data are available from the corresponding author upon reasonable request.

2) Yields need to be provided for every reaction (*Ru XP, Os XP, Os X2*).

Response 17:

The yields of all organic reactions are added.

3) *Full reaction procedures and reagent quantities need to be provided for all reactions. Simply saying the “target molecule was obtained by using the similar protocol of synthesis...” is not acceptable for a synthetic paper.*

Response 18:

The reaction procedures of Fe XY, Co XY, Ru XP, Os XP and Os X₂ are rewritten with details. Reagent quantities of the syntheses of Os XY and Ir XY are added.

4) *Significant figures are all over the place and need to be edited to be consistent.*

Response 19:

We have edited the significant figures to be consistent for organic synthesis. Owing to many types of numbers for partial supporting information before organic synthesis, we have modified the significant figures in this part.

5) *Check for typos. For example, line 371, it should be (NH₄)₂OsCl₆, not “(NH₄)OsCl₆”*

Response 20:

(NH₄)OsCl₆ was replaced by (NH₄)₂OsCl₆.

6) *Figure S33 and S34 should have cleaner NMRs. These compounds are too simple to have such visible impurities.*

Response 21:

These compounds were purified, and NMRs were updated as new Figure S33 and S36.

7) *ALL small molecules need at minimum LRMS characterization, if not HRMS. There's no reason to not have this simple characterization.*

Response 22:

We have added mass spectra for all small molecules.

Carefully check grammar in the Supporting Information. For example...

Line 71, “appears” should be “appear”

Line 72, “there is” should be “there are”

Line 73, “within [the] self-assembled monolayer”

Response 23:

These words were modified.

Line 74, the CVs confirm the completion of the self-coupling reactions of the carbazolyls. It doesn't make sense for reactions to be found in CVs.

Response 24:

We have modified the sentences. Ref. S1-c was deleted.

Figure S1. Successive CVs of self-assembled Os^{II}PX (a) and Ru^{II}PX (b) on ITO coated glasses at 100 mV/s in 0.1 M Bu₄NClO₄/MeCN monomer free electrolyte. Strong irreversible oxidation peaks of carbazolyl appear at 0.95~0.98 V, and the redox peaks of 3,3'-bicarbazolyls at 0.50~0.60 V^{S1} cannot be observed, implying that there are no self-coupling reactions of carbazolyls within the self-assembled monolayer. Usually, the redox peaks of 3,3'-bicarbazolyls at 0.5~0.6 V can be clearly observed (Figures in Ref. S1 and Ref. S2).

Figure 2. Cyclic voltammogram of the ITO substrate modified with a carbazole monolayer. TBAH was used as the supporting electrolyte. Scan rate: 20 mV/s.

Ref. S1

Figure 1. CV curves of C₂Th SAM-modified gold electrode in acetonitrile using Bu₄NPF₆ as supporting electrolyte. Scan rate of 20 mV s⁻¹, scan range of 0.00 V – 0.95 V. The integration of shaded area is used to calculate the polymerization percent and the surface coverage of SAM.

Ref. S2

Line 83, “band[s]”

Line 85, “[Moderate] absorption bands [in] the range of 490-600 nm”

...Etc.

Response 25: We have modified these sentences.

Reviewer #2 (Remarks to the Author):

The authors have made significant changes, and answer my questions.

Therefore i confirm my previous recommendation: publication.

Reviewer #3 (Remarks to the Author):

The authors have addressed all of my concerns, I recommend this for publication.

REVIEWERS' COMMENTS:

Reviewer #1 (Remarks to the Author):

The authors have addressed my concerns, I recommend this for publication.

Reviewer #2 (Remarks to the Author):

i think due to the amount of additional work, i recommend publication